# Oxidative 1,2-carboamination of alkenes with alkyl nitriles and amines toward $\gamma$-amino alkyl nitriles

Yan-Yun Liu[1,2,*], Xu-Heng Yang[1,2,*], Ren-Jie Song[1,2], Shenglian Luo[1,2] & Jin-Heng Li[1,2,3]

Difunctionalization of alkenes has become a powerful tool for quickly increasing molecular complexity in synthesis. Despite significant progress in the area of alkene difunctionalization involving the incorporation of a nitrogen atom across the C–C double bonds, approaches for the direct 1,2-carboamination of alkenes to produce linear $N$-containing molecules are scarce and remain a formidable challenge. Here we describe a radical-mediated oxidative inter-molecular 1,2-alkylamination of alkenes with alkyl nitriles and amines involving $C(sp^3)$–H oxidative functionalization catalysed by a combination of $Ag_2CO_3$ with iron Lewis acids. This three-component alkene 1,2-alkylamination method is initiated by the $C(sp^3)$–H oxidative radical functionalization, which enables one-step formation of two new chemical bonds, a C–C bond and a C–N bond, to selectively produce $\gamma$-amino alkyl nitriles.

[1] Key Laboratory of Jiangxi Province for Persistent Pollutants Control and Resources Recycle, Nanchang Hangkong University, Nanchang 330063, China. [2] State Key Laboratory of Chemo/Biosensing and Chemometrics, Hunan University, Changsha 410082, China. [3] State Key Laboratory of Applied Organic Chemistry, Lanzhou University, Lanzhou 730000, China. * These authors contributed equally to this work. Correspondence and requests for materials should be addressed to S.L. (email: sllou@hnu.edu.cn) or to J.-H.L. (email: jhli@hnu.edu.cn).

**D**ifunctionalization of alkenes represents one of the most powerful and straightforward tools to build complex molecules via one-step construction of two chemical bonds that possess significantly synthetic utility in chemical synthesis[1–6]. One of the major synthetic targets for such transformations, including diamination[7–15], aminooxygenation[16–24], aminohalogenation[25–30] and carboamination[31–36], is the incorporation of a nitrogen atom (amino, amide or azide groups) across the C–C double bonds to build useful N-containing molecules through the formation of a C–N bond. Despite significant progress in the field, approaches of the alkene carboamination for producing linear N-containing molecules are scarce and remain a great challenge (Fig. 1a): available intermolecular transformations for producing linear N-containing molecules are restricted to the special amination reagents[33–36]. Further, to our knowledge, three-component carboamination reactions of the alkenes via C–H functionalization have never been reported.

In recent years, the C–H oxidative functionalization reaction has attracted much attention due to its inherent features, such as high step economy and atom economy[1–6,37–41]. Typical transformations include the difunctionalization of alkenes with alkyl C($sp^3$)–H bonds[42–55] and the majority of which rely on the formation of a $sp^3$-hybridized carbon-centred radical from the oxidative cleavage of the corresponding alkyl C($sp^3$)–H bond followed by addition across the C–C double bond[43–59]. However, such approaches are restricted to the 1,2-arylalkylation[52–63], 1,2-dialkylation[54] and 1,2-oxyalkylation[55–59] of the alkenes, and the available three-component transformations are scarce[53,55]. In light of these literature results[43–59] and our continuous interest in oxidative radical reactions[60–63], we envisioned that this C–H oxidative radical functionalization strategy might be viable to achieve 1,2-carboamination of alkenes with new-conceptual, general and straightforward features.

Herein, we report an iron-catalysed oxidative three-component 1,2-carboamination of alkenes with alkyl nitriles and amines through C($sp^3$)–H oxidative radical functionalization to assemble γ-amino alkyl nitriles using Ag$_2$CO$_3$ as oxidant (Fig. 1b). The reaction enables the simultaneous formation of two new chemical bonds, a C–C bond and a C–N bond, by a sequence of C–H oxidative cleavage, radical addition across the alkenes and amination in a highly atom-economic and selective manner[64–68].

## Results

**Reaction optimization**. We initiated the study by investigating various reaction parameters for the three-component reaction of *p*-methoxystyrene (**1a**) with acetonitrile (**2a**) and dibenzylamine

(**3a**) (Table 1). A combination of 10 mol% Fe(OTf)$_3$, 2 equiv Ag$_2$CO$_3$, 120 °C and 24 h were found as the optimal reaction conditions for the conversion of alkene **1a**, nitrile **2a** and amine **3a** to the desired product **4** in 82% yield (entry 1). The results suggest that Ag$_2$CO$_3$ is the real catalysts and Fe(OTf)$_3$ only serves as a Lewis acid to promote the reaction (entries 2 and 3): although in the absence of Fe(OTf)$_3$ transformation of alkene **2a** to **4** took place albeit giving a lower yield (entry 2), no desired reaction was observed without Ag salts (entry 3). Other Ag salts, including Ag$_2$O, AgOAc and AgNO$_3$, had the catalytic activity for the reaction, but they were less effective than Ag$_2$CO$_3$ (entries 4–6). Among the amount of Ag$_2$CO$_3$ examined, the use of 2 equiv was turned out to be preferred (entries 1, 7 and 8). Encouraged by these, a series of other Lewis acids, such as FeCl$_3$, Yb(OTf)$_3$, Cu(OTf)$_2$ and In(OTf)$_3$, were tested (entries 9–12): they could improve the reaction, but were less effective than Fe(OTf)$_3$. Notably, the use of other bases, Na$_2$CO$_3$ or Cs$_2$CO$_3$, instead of Ag$_2$CO$_3$, resulted in no formation of product **4** (entries 13 and 14), suggesting that Ag$_2$CO$_3$ may act as an oxidant and a catalyst, not a base. Notably, the reported efficient oxidative systems, *t*BuOO*t*Bu di-*tert*-butyl peroxide (DTBP)[42–55] or Ag$_2$CO$_3$/K$_2$S$_2$O$_8$ (refs 60–67) displayed rather lower activity for the reaction (entries 15 and 16). We found that the reaction was sensitive to the temperatures (entries 17 and 18): a lower temperature (100 °C) had a negative effect on the reaction, whereas a higher temperature (130 °C) did not improve the yield compared with the results at 120 °C. Gratifyingly, the reaction could be successfully performed in PhCF$_3$ medium (entry 19).

**Substrate scope with amines and amides**. We next explored the scope of this Ag$_2$CO$_3$-mediated 1,2-carboamination protocol under the optimal reaction conditions with regard to alkenes **1**, nitriles **2** and amines **3** (Tables 2 and 3). We first turned our attention to investigate the applicability of the optimal conditions in the reaction with various amines **3b–m** in the presence of alkene **1a** and acetonitrile **2a** (Table 2). The resulted indicated that a wide range of secondary and primary amines **3b–j** were smoothly converted to the desired products **5–13** in moderate to good yields. *N*-Methyl-1-phenylmethanamine (**3b**) was viable to furnish **5** with 89% yield in the presence of Fe(OTf)$_3$ and Ag$_2$CO$_3$. For other amines **3c–j**, however, Fe(OTf)$_3$ displayed less efficient than FeCl$_3$ (products **6–13**): although treatment of alkene **1a** with nitrile **2a**, diisopropylamine (**3c**), Fe(OTf)$_3$ and Ag$_2$CO$_3$ afforded **6** in 56% yield, the use of FeCl$_3$ instead of Fe(OTf)$_3$ enhanced the yield to 66%. Similarly, the yield of **10** from the reaction with morpholine (**3g**) increased from 74 to 83% when using FeCl$_3$ instead of Fe(OTf)$_3$. To our delight, the optimal

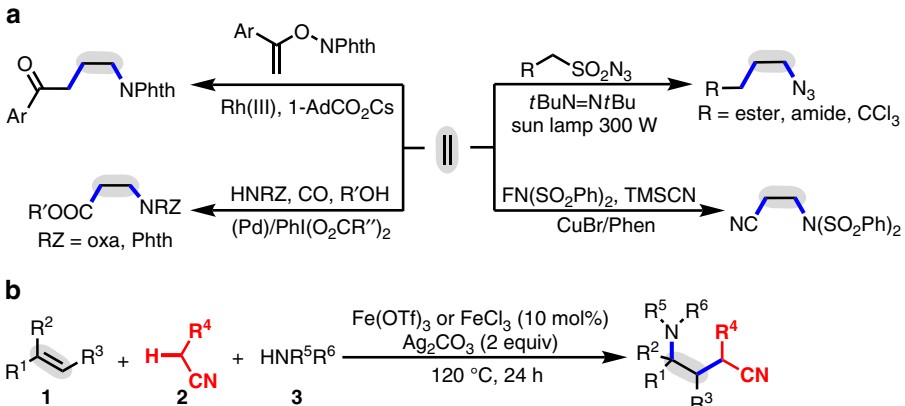

**Figure 1 | 1,2-Carboamination of alkenes.** (**a**) Previous work for 1,2-carboamination of alkenes[33–36]. (**b**) Our radical-mediated three-component, oxidative carboamination between alkenes, alkyl nitriles and amines using a C–H oxidative functionalization.

## Table 1 | Screening of optimal reaction conditions.

| Entry | Variation from the optimal conditions | Yield (%)* |
|---|---|---|
| 1 | None | 82 |
| 2 | Without Fe(OTf)₃ | 33 |
| 3 | Without Ag₂CO₃ | 0 |
| 4 | Ag₂O instead of Ag₂CO₃ | 34 |
| 5 | AgOAc instead of Ag₂CO₃ | 15 |
| 6 | AgNO₃ instead of Ag₂CO₃ | 6 |
| 7 | Ag₂CO₃ (1 equiv) | 53 |
| 8 | Ag₂CO₃ (3 equiv) | 80 |
| 9 | FeCl₃ instead of Fe(OTf)₃ | 68 |
| 10 | Yb(OTf)₃ instead of Fe(OTf)₃ | 68 |
| 11 | Cu(OTf)₂ instead of Fe(OTf)₃ | 41 |
| 12 | In(OTf)₃ instead of Fe(OTf)₃ | 63 |
| 13 | Na₂CO₃ instead of Ag₂CO₃ | 0 |
| 14 | Cs₂CO₃ instead of Ag₂CO₃ | 0 |
| 15 | (tBuO)₂ instead of Ag₂CO₃ | trace |
| 16† | Ag₂CO₃/K₂S₂O₈ instead of Ag₂CO₃ | trace |
| 17 | At 100 °C | 52 |
| 18 | At 130 °C | 80 |
| 19‡ | MeCN (0.2 ml; 3.85 mmol) | 78 |

Experiments were performed with **1a** (0.3 mmol), MeCN **2a** (2 ml), Bn₂NH **3a** (2 equiv), Fe(OTf)₃ (10 mol%), Ag₂CO₃ (2 equiv), 120 °C and 24 h.
*Average isolated yield twice.
†Ag₂CO₃ (20 mol%) and K₂S₂O₈ (2 equiv).
‡PhCF₃ (1.8 ml).

## Table 3 | Variation of the alkenes (1).

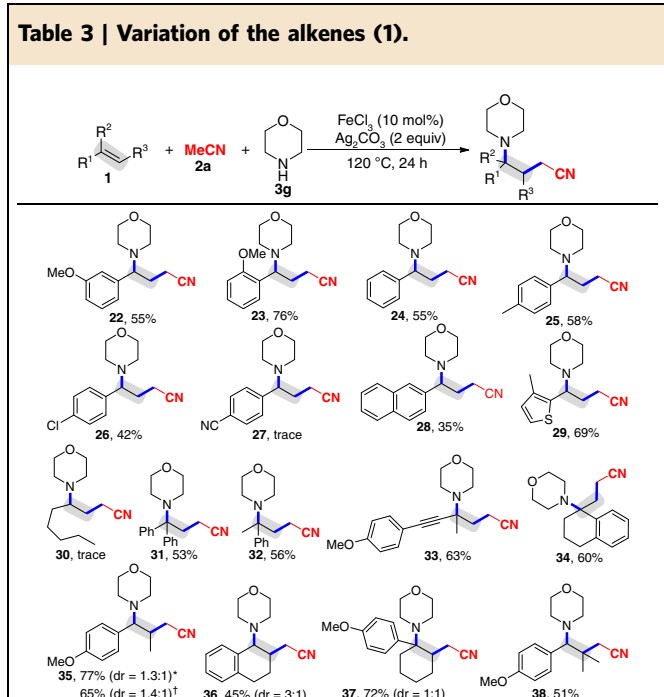

Experiments were performed with **1** (0.3 mmol), nitrile **2a** (2 ml), amine **3g** (2 equiv), FeCl₃ (10 mol%), Ag₂CO₃ (2 equiv), 120 °C, 24 h. The dr value is given in parentheses as determined by ¹H NMR analysis of the crude product.
*Using (E)-1-methoxy-4-(prop-1-en-1-yl)benzene (**1o**).
†Using (Z)-1-methoxy-4-(prop-1-en-1-yl)benzene (**1p**).

## Table 2 | Variation of the alkyl nitriles (2) and amines (3).

Experiments were performed with **1a** (0.3 mmol), nitrile **2** (2 ml), amine **3** (2 equiv), [Fe] (10 mol%), Ag₂CO₃ (2 equiv), 120 °C, 24 h. The dr value is given in parentheses as determined by ¹H NMR analysis of the crude product.
*Using Fe(OTf)₃.
†Using FeCl₃.

conditions were compatible with sulfonamides **3e**, **3k** and **3l**, giving products **8**, **14** and **15** in high yields. Unfortunately, attempt to difunctionalization with acetamide **3m** failed to build 1,2-carboamination product **16**.

Subsequently, the scope of alkyl nitriles **2** was exploited in the presence of alkene **1a**, morpholine **3g**, FeCl₃ and Ag₂CO₃ (Table 2). In the case of butyronitrile **2b**, the reaction afforded **17** in 66% yield. Gratifyingly, the reaction was well tolerated of various acetonitriles **2c–e** bearing a Ph group, a MeO group or a CO₂Et group at the α position, generating **18–20** in 50–70% yields. An interesting observation was that secondary alkyl nitrile **2f** containing a cyclohexyl ring also proceeded the reaction and resulted in the formation of **21** in 58% yield.

**Substrate scope with alkenes**. The optimal conditions were applicable to an array of alkenes **1b–f**, **1h–i** and **1k–s** (products **22–26**, **28–29** and **31–38**), but electron-withdrawing aryl alkene **1g** and simple aliphatic alkene, namely oct-1-ene (**1j**), had no reactivity (products **27** and **30**; Table 3). Initially, the substitution effect of the aryl ring at the terminal alkenes were examined: several substituted aryl rings, such as m-MeOC₆H₄, o-MeOC₆H₄, C₆H₅, p-MeC₆H₄, m-MeC₆H₄, naphthalen-2-yl and 3-methylthiophen-2-yl, were perfectly tolerated, and both the electronic nature of the aryl group and the substituent position on the aryl group had an impact on the reactivity (products **22–29**). Using m-methoxystyrene (**1b**), for example, afforded **22** in 55% yield, whereas bulky o-methoxystyrene (**1c**) furnished **23** in 76% yield. Alkene **1f** having a weak electron-deficient 4-ClC₆H₄ group successfully underwent the 1,2-alkylamination reaction to offer **26**, albeit in a diminished yield. However, alkene **1g** having a strong electron-deficient 4-CNC₆H₄ group had no reactivity (product **27**). Gratifyingly, the optimal conditions were consistent with 1,1-disubstituted alkenes, including 1,1-diphenylethylene (**1k**), prop-1-en-2-ylbenzene (**1l**), 1-methoxy-4-(3-methylbut-3-en-1-yn-1-yl)benzene (**1m**) and 1-methylene-1,2,3,4-tetrahydronaphthalene (**1n**), generating **31–34** with concomitant formation of a quaternary carbon centre. A particularly attractive feature of this 1,2-alkylamination is the ability to enable the conversion of

**Figure 2 | Control experiments and utilizations of product 4.** (**a**) Radical testing experiment based on the selectivity. (**b**) Trapping experiments with a stoichiometric amount of radical inhibitors. (**c**) Kinetic isotopic effect (KIE) study. (**d**) Synthetic utilizations.

**Figure 3 | Possible mechanism.** The alkyl radical **C** is generated from decomposition of the AgCH₂CN intermediate **B** via single-electron transfer. Subsequently, addition of the alkyl radical **C** across the C–C double and oxidative amination afford product **4**.

di- and trisubstituted internal alkenes **1o–s** to diverse complex products **35–38** in moderate to good yields. It was noted that the reaction of (*E*)-1-methoxy-4-(prop-1-en-1-yl)benzene (**1o**) or (*Z*)-1-methoxy-4-(prop-1-en-1-yl)benzene (**1p**) had no retention of geometrical selectivity in the double bond (product **35**), which supported a radical process.

**Control experiments and mechanistic studies**. Using (1-cyclo-propylvinyl)benzene (**1t**) to react with nitrile **2a** and amine **3g**, the 1,2-alkylarmination product **39** along with the mono alkylation/ring-opening/cyclization product **40** was observed (Fig. 2a)[55]. Notably, the reaction of alkene **1a** with nitrile **2a** and amine **3a** could not take place in the presence of a stoichiometric amount of radical

inhibitors, such as 2,2,6,6-Tetramethyl-1-piperidinyloxy (TEMPO), 2,6-di-*tert*-butyl-4-methylphenol and hydroquinone (Fig. 2b).

In addition, under the optimal conditions nitrile **2a** reacted with TEMPO afforded product **41**. These results suggested that the current reaction is triggered by a free-radical process. The kinetic isotope effect experiment gave a large kinetic isotope effect value ($k_H/k_D = 2.7$), implying that the cleavage of the $C(sp^3)$–H bond may be rate-limiting (Fig. 2c and for the detailed information, see Supplementary Fig. 39)[37–55]. Gratifyingly, product **4** were easily converted to 1,4-diamine **42**, $\gamma$-amino acid **43** and $\gamma$-amino amide **44** in good yields (Fig. 2d)[69,70].

Consequently, the mechanisms for the Ag₂CO₃-mediated 1,2-alkylamination reaction was proposed (Fig. 3)[31–68].

Coordination of the nitrogen atom in MeCN **2a** with AgCO$_3$ gives the intermediate **A**, which sequentially reacts with AgCO$_3$ to afford the AgCH$_2$CN intermediate **B** and AgHCO$_3$. The decomposition the AgCH$_2$CN intermediate **B** readily takes place under heating to form the alkyl radical **C** (supported by the results of Fig. 2b), AgHCO$_3$ and the Ag$^0$ species [Ag(s)] through single electron transfer[42–65]. Subsequently, addition of the alkyl radical **C** across the C–C double bond in alkene **1a** produces the alkyl radical intermediate **D** (supported by the reaction of alkene **1t**; Fig. 2a). Intermediate **D** is converted into the carbon-centered cation **E**, followed by reaction with amine **3a** affords the product **4**, AgHCO$_3$ and the Ag$^0$ species through a sequence of oxidation and nucleophilic addition[64–68]. Notably, the radical intermediates **C** and **D** can be stabilized by Lewis acids, thus improving the yields.

In summary, we have developed a silver-mediated intermolecular 1,2-alkylamination of alkenes with alkyl nitriles and amines involving C(sp$^3$)–H oxidative radical functionalization for producing γ-amino alkyl nitriles. The generality of such an intermolecular 1,2-alkylamination reaction is demonstrated by a wide scope with respect to alkenes, alkyl nitriles and amines. The radical mechanism was also discussed according to the control experiments. Importantly, applications of the products, γ-amino alkyl nitriles, to prepare other valuable synthons have been examined. Currently, our laboratory is working to apply this C–H oxidative radical functionalization strategy in synthesis.

## Methods

**General procedure for 1,2-carboamination of alkenes.** To a Schlenk tube were added Fe(OTf)$_3$ or FeCl$_3$ (10 mol%), Ag$_2$CO$_3$ (0.6 mmol), alkene **1** (0.3 mmol), amine **2** (0.6 mmol) and MeCN (2 ml). Then the tube was recharged with argon and the mixture was stirred at 120 °C for 24 h. After cooling to room temperature, the mixture was filtered through a small plug of silica gel to remove the precipitate and washed with with EtOAc (3 × 10 ml). The solvent was then removed in vacuo and the residue was further purified by silica gel flash column chromatography (10–40% ethyl acetate/hexane + 0.1% Et$_3$N) to afford the desired product.

**Data availability.** The X-ray crystallographic coordinates for structures reported in this study have been deposited at the Cambridge Crystallographic Data Centre under deposition number 1453224 (**4**). These data can be obtained free of charge from The Cambridge Crystallographic Data Centre via www.ccdc.cam.ac.uk/data_request/cif. All other data supporting the findings of this study are available within the article and its Supplementary Information file or from the authors upon reasonable request. For NMR spectra of the compounds in this article, see Supplementary Figs 1–39.

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

## Acknowledgements

We thank the Natural Science Foundation of China (numbers 21472039 and 21625203) and the Hunan Provincial Natural Science Foundation of China (number 13JJ2018).

## Author contributions

Y.-Y.L. and X.-H.Y. contributed equally to this work. Y.-Y.L., X.-H.Y., S.L. and J.-H.L. conceived the project and wrote the manuscript. Y.-Y.L. and X.-H.Y performed the experiments. Y.-Y.L., X.-H.Y. and R.-J.S. analysed the data.

## Additional information

**Competing financial interests:** The authors declare no competing financial interests.

**Publisher's note**: 

