## [Peer review file · Nature Communications]

Reviewers' comments:

Reviewer #1 (Remarks to the Author):

Li and Luo disclosed an interesting oxidative carbonamination of alkenes, in which alkyl nitriles were used as carbon source. In this transformation, silver salt is crucial for the C-H cleavage to generate alkyl carbon radical, which can react with styrene to generate benzylic radical. Further reaction with amine, amino-alkyl nitriles were obtained efficiently. As mentioned in the manuscript, broad substrate scope of the amines and alkyl nitriles was observed. For the styrenes, electron-rich alkenes were good for the three component coupling. Furthermore, the preliminary mechanistic study is consistent with a radical pathway. Overall, this manuscript is quite interesting, and should be more useful in the organic synthesis. Thus, I recommend to publish this work in *Nature Commun.* with minor revision:

(1) Entries 9, do the reactions give the same yields in the different iron or Yb catalyst? However, only one yield was given in the item, which might be incorrect? Please recheck. If this is not true, what's the really result with Yb(OTf)₃? What's the difference between iron catalyst and Yb catalyst?

(2) For the mechanism part, the C-H cleavage was proposed as rate-determining step, but no detailed information was provided. In this reaction, 2 equivalent of Ag₂CO₃ was required. In the previous reported literature (ref. 8a), AgCH₂CN was proposed to form initially. Thus, the alkyl radical might be derived from AgCH₂CN. The similar work on the Ag-mediated alkylation of alkene from acetonitriles (*Chin. J. Chem.* 2014, 32, 681-684) should be cited.

(3) For the final C-N bond formation, Lei and coworkers proposed that an oxidation of benzylic radical to carbon cation (*Chem. Asian J.* 2015, 10, 96 – 99 and *Adv. Synth. Catal.* 2014, 356, 2873 – 2877). Thus, the same pathway is possibly involved in the reaction, and these references should be cited.

Reviewer #2 (Remarks to the Author):

Luo, Li and co-workers report in this paper an oxidative 1,2-carboamination of alkenes. Reaction of alkenes with alkyl nitriles and amines in the presence of Ag₂CO₃ (2.0 equiv) and a catalytic amount of Lewis acid afforded gamma-amino alkyl nitriles in good yields. This is an interesting transformation and I recommend its publication in *Nat Commun* after minor revision noted below:

a) P3, scheme 3. The equation discussing the generation of acetonitrile radical A is a little bit too simplified. Actually, I did not understand how the radical A was generated. Did the SET process occur directly between Ag₂CO₃ and MeCN, or it occurred via the alpha-carbanion of the acetonitrile. In the second case, how the anion was generated? This would be very informative to readers who are interested in this type of reactions.

b) Recent papers dealing with the cyanoalkylative difunctionalization of alkenes are pertinent to the work described in this paper (*ACIE*, 2016, 55, 9249; *Chem Commun.* 2016, 52, 11100).

Reviewers' Comments:

Reviewer #1 (Remarks to the Author):

Li and Luo disclosed a interesting oxidative carbonamination of alkenes, in which alkyl nitriles was used as carbon source. In this transformation, silver salt is crucial for the C-H cleavage to generate alkyl carbon radical, which can react to styrene to generate benzylic radical. Further react with amine, amino-alkylnitriles were obtained efficiently. As mentioned in the manuscript, broad substrate scope of the amines and alkyl nitriles was observed. For the styrenes, electron-rich alkenes were good for the three component coupling. Furthermore, the preliminary mechanistic study is consistent with a radical pathway. Overall, this manuscript is quite interesting, and should be more useful in the organic synthesis. Thus, I recommend to publish this work in Nature Commun. with minor revision:

- (1) Entries 9, do the reactions give the same yields in the different iron or Yb catalyst? However, only one yield was given in the item, which might be incorrect? Please recheck. If this is not true, what's the really result with Yb(OTf)₃? What's the different between iron catalyst and Yb catalyst?
- (2) For the mechanism part, the C-H cleavage was proposed as rate-determining step, but no detailed information was provided. In this reaction, 2 equivalent of Ag₂CO₃ was required. In the previous reported literature (ref. 8a), AgCH₂CN was proposed to form initially. Thus, the alkyl radical might be derived from AgCH₂CN. The similar work on the Ag-mediated alkylation of alkene from acetonitriles (Chin. J. Chem. 2014, 32, 681-684) should be cited.
- (3) For the final C-N bond formation, Lei and cowork proposed that a oxidation of benzylic radical to carbon cation (Chem. Asian J. 2015, 10, 96 – 99 and Adv. Synth. Catal. 2014, 356, 2873 – 2877). Thus, the same pathway is possibly involved in the reaction, and these reference should be cited.

Our Revision:

Thanks,

Comments: (1) Entries 9, do the reactions give the same yields in the different iron or Yb catalyst? However, only one yield was given in the item, which might be incorrect? Please recheck. If this is not true, what's the really result with Yb(OTf)₃? What's the different between iron catalyst and Yb catalyst?

Our response: Yes, identical yields were obtained when using the reactions the different iron or Yb catalyst. We have rechecked all the results in Table 1. The results showed that both FeCl₃ and Yb(OTf)₃ had the same catalytic activity because they acted as the nature of Lewis acids (See Table 1 and the text).

Comments: (2) For the mechanism part, the C-H cleavage was proposed as rate-determining step, but no detailed information was provided. In this reaction, 2 equivalent of Ag_2CO_3 was required. In the previous reported literature (ref. 8a), AgCH_2CN was proposed to form initially. Thus, the alkyl radical might be derived from AgCH_2CN . The similar work on the Ag-mediated alkylation of alkene from acetonitriles (*Chin. J. Chem.* 2014, 32, 681-684) should be cited.

Our response: 1) For the mechanism part, the C-H cleavage was proposed as rate-determining step, which is supported by a large KIE value ($k_H/k_D = 2.7$). The detailed information was provided in page S58 of Supporting Information.

We have revised the mechanism according to the reviewer's comments (see the text, Scheme 3) and the literature on the Ag-mediated alkylation of alkene from acetonitriles (*Chin. J. Chem.* 2014, 32, 681-684) was cited (see reference 83).

Comments: (3) For the final C-N bond formation, Lei and coworker proposed that a oxidation of benzylic radical to carbon cation (*Chem. Asian J.* 2015, 10, 96 – 99 and *Adv. Synth. Catal.* 2014, 356, 2873 – 2877). Thus, the same pathway is possibly involved in the reaction, and these reference should be cited.

Our response: The relative literatures (*Chem. Asian J.* 2015, 10, 96 – 99 and *Adv. Synth. Catal.* 2014, 356, 2873 – 2877) were cited (see references 86 and 87).

Reviewer #2 (Remarks to the Author):

Luo, Li and co-workers report in this paper an oxidative 1,2-carboamination of alkenes. Reaction of alkenes with alkylnitriles and amines in the presence of Ag_2CO_3 (2.0 equiv) and a catalytic amount of Lewis acid afforded gamma-amino alkylnitriles in good yields. This is an interesting transformation and I recommend its publication in *Nat Commun* after minor revision noted below:

a) P3, scheme 3. The equation discussing the generation of acetonitrile radical A is a little bit too simplified. Actually, I did not understand how the radical A was generated. Did the SET process occurred directly between Ag_2CO_3 and MeCN, or it occurred via the alpha-carbanion of the

acetonitrile. In the second case, how the anion was generated? This would be very informative to readers who are interested in this type of reactions.

b) Recent papers dealing with the cyanoalkylative difunctionalization of alkenes are pertinent to the work described in this paper (ACIE, 2016, 55, 9249; Chem Commun. 2016, 52, 11100).

Our Revision:

Thanks,

Comments: a) P3, scheme 3. The equation discussing the generation of acetonitrile radical A is a little bit too simplified. Actually, I did not understand how the radical A was generated. Did the SET process occurred directly between Ag_2CO_3 and MeCN, or it occurred via the alpha-carbanion of the acetonitrile. In the second case, how the anion was generated? This would be very informative to readers who are interested in this type of reactions.

Our response: The possible mechanism for the reaction was revised according to the reviewer suggestion: we have discussed how the radical A was generated and how the anion was generated (See Scheme 3 in the text), which was proposed by many literatures on the Ag-mediated dialkylation of alkenes with acetonitriles (see references 70-89).

Comments:

b) Recent papers dealing with the cyanoalkylative difunctionalization of alkenes are pertinent to the work described in this paper (ACIE, 2016, 55, 9249; Chem Commun. 2016, 52, 11100).

Our response: Recent papers (*Angew. Chem. Int. Ed.* **2016**, 55, 9249; *Chem Commun.* **2016**, 52, 11100) were cited as references 88 and 89.

REVIEWERS' COMMENTS:**Reviewer #1 (Remarks to the Author):**

Li and coauthors have well addressed all of my questions, and the revised manuscript is significantly improved. This referee believe this is a excellent works, which is deserved to publish in Nature Communications.